# Low-Dose Acetylsalicylic Acid Treatment in Non-Skull-Base Meningiomas: Impact on Tumor Proliferation and Seizure Burden

**DOI:** 10.3390/cancers14174285

**Published:** 2022-09-01

**Authors:** Johannes Wach, Ági Güresir, Hartmut Vatter, Ulrich Herrlinger, Albert Becker, Marieta Toma, Michael Hölzel, Erdem Güresir

**Affiliations:** 1Department of Neurosurgery, University Hospital Bonn, 53127 Bonn, Germany; 2Division of Clinical Neurooncology, Department of Neurology and Centre of Integrated Oncology, University Hospital Bonn, 53127 Bonn, Germany; 3Department of Neuropathology, University Hospital Bonn, 53127 Bonn, Germany; 4Institute of Pathology, University Hospital Bonn, 53127 Bonn, Germany; 5Institute of Experimental Oncology, University Hospital Bonn, 53127 Bonn, Germany

**Keywords:** acetylsalicylic acid, aspirin, meningioma, proliferation, seizure

## Abstract

**Simple Summary:**

MIB-1 index is an established immunohistochemical marker reflecting the proliferative potential in meningiomas. Cyclooxygenase-2 has been demonstrated to be positively correlated with MIB-1 index in meningiomas. In a recent investigation, we revealed that cranial non-skull-base meningiomas have increased proliferative potential and increased inflammatory burden compared to other anatomical locations of meningiomas. The role of nonsteroidal anti-inflammatory drugs (NSAIDs) regarding proliferative activity and location-specific symptoms in non-skull-base meningiomas is unclear so far. In the present investigation, we therefore analyzed the impact of low-dose (100 mg) acetylsalicylic acid (ASA) treatment on the MIB-1 index and baseline symptomatic epilepsy. We identified that elderly female non-skull-base meningioma patients with ASA intake had significantly lower MIB-1 indices. Furthermore, ASA intake significantly reduced the preoperative seizure burden in non-skull-base meningioma. We believe that further research on NSAID treatment in non-skull-base meningiomas is necessary to enhance a tailored treatment scheduling.

**Abstract:**

MIB-1 index is an important predictor of meningioma progression and was found to be correlated with COX-2 expression. However, the impact of low-dose acetylsalicylic acid (ASA) on MIB-1 index and clinical symptoms is unclear. Between 2009 and 2022, 710 patients with clinical data, tumor-imaging data, inflammatory laboratory (plasma fibrinogen, serum C-reactive protein) data, and neuropathological reports underwent surgery for primary cranial WHO grade 1 and 2 meningioma. ASA intake was found to be significantly associated with a low MIB-1 labeling index in female patients ≥ 60 years. Multivariable analysis demonstrated that female patients ≥ 60 years with a non-skull-base meningioma taking ASA had a significantly lower MIB-1 index (OR: 2.6, 95%: 1.0–6.6, *p* = 0.04). Furthermore, the intake of ASA was independently associated with a reduced burden of symptomatic epilepsy at presentation in non-skull-base meningiomas in both genders (OR: 3.8, 95%CI: 1.3–10.6, *p* = 0.03). ASA intake might have an anti-proliferative effect in the subgroup of elderly female patients with non-skull-base meningiomas. Furthermore, anti-inflammatory therapy seems to reduce the burden of symptomatic epilepsy in non-skull-base meningiomas. Further research is needed to investigate the role of anti-inflammatory therapy in non-skull-base meningiomas.

## 1. Introduction

Meningiomas represent 36.4% of all central nervous system (CNS) tumors and are predominantly benign neoplasms arising from the arachnoid cap cells and dural border cells [1,2]. Maximum cytoreductive surgery is suggested as the mainstay therapy for symptomatic meningiomas, and, from a histopathological point of view, World Health Organization (WHO) grades 1 and 2 account for approximately 97% of all meningiomas [3,4]. Therefore, the current standard of care remains surgery followed by radiation therapy in partially resected atypical and all anaplastic meningiomas. To date, chemotherapy, hormonal therapy, and immunotherapeutic approaches have demonstrated limited benefits and are predominantly intended for anaplastic meningiomas [5]. 

Cellular proliferative potential is a known mechanism of oncogenesis, and the Molecular Immunology Borstel 1 (MIB-1) labeling index is an immunohistochemical method to detect the Ki-67 antigen, which is exclusively present in cell nuclei during the G1, S, and G2 phases of the cell division cycle [6]. Therefore, this method enables the identification of the growing fraction of neoplastic cell areas [7,8,9]. Several investigations have demonstrated that an increased MIB-1/Ki-67 labeling index is an independent risk factor for meningioma progression [10,11,12,13]. 

Chronic inflammation has been identified as an emerging source of tumorigenesis in various neoplasms. One of the main avenues debated is the cyclooxygenase (COX) group, which includes enzymes catalyzing the production of prostaglandins, which are a major driver of inflammatory burden [14]. Acetylsalicylic acid has been assumed as a protective drug for several types of inflammation-associated cancer [15,16].

This pathophysiological mechanism was already previously translated to the evolution and biology of meningioma. Kato et al. [17] showed the tumorigenic activity of COX-2 expression in 76 human cases of cranial meningioma, which resulted in a significant correlation between COX-2 expression and MIB-1 labeling index. Moreover, COX-2 inhibition was observed to inhibit meningioma growth and induce apoptosis in an in vitro treatment setting [18]. Furthermore, there are emerging data suggesting a potential therapeutic strategy for epilepsy management involving COX-2 inhibition [19]. Management of symptomatic epilepsy is of paramount importance in this disease and affects approximately 26% to 34% of all non-skull-base meningioma (falcine, convexity, parasagittal) patients at presentation [20,21]. To date, there is no substantial evidence from a prospective randomized phase II trial supporting the proof of concept or clinical benefit of COX-2 inhibition in human cranial meningiomas. 

The present study investigates our patient cohort of sporadic primary cranial WHO grade 1 and 2 meningiomas regarding the impact of ASA intake on proliferation in cranial meningiomas and burden of baseline symptomatic epilepsy in non-skull-base meningiomas.

## 2. Materials and Methods

### 2.1. Patient Population

This investigation reviewed 1035 consecutive sporadic cranial meningioma patients who underwent surgery between January 2009 and January 2022. The aim of the present single-center series is focused on the impact of ASA intake in primary sporadic cranial meningiomas on proliferative potential reflected by the MIB-1 labeling index and symptomatic epilepsy as a location-specific symptom in non-skull-base meningiomas. Patients with a recurrent meningioma, with prior radiotherapy, and neurofibromatosis type 2 patients were excluded because of their different clinical symptoms, neuropathology, and treatment strategies [13,22,23]. Furthermore, patients without immunohistochemical data depicting the MIB-1 labeling index, without documentation of the preoperative prescribed regular medication, as well as patients without preoperative plasma (fibrinogen) and serum (C-reactive protein) values were excluded. A total of 707 patients were included in the final study cohort.

### 2.2. Data Recording and Radiological Features

Clinical patient data such as age, sex, comorbidities, Karnofsky performance status (KPS), body mass index (BMI), length of stay (in days), presence of seizure, prior regular medication including acetylsalicylic acid (daily dose of 100 mg), intake of non-steroidal anti-inflammatory drugs or cortisol, American society of anesthesiologists physical status classification system, WHO grading, immunohistochemical examinations, and peritumoral edema were recorded in a computerized database (SPSS, version 27 for Windows, IBM Corp., Armonk, NY, USA). Magnetic resonance imaging (MRI) was routinely performed within 48 h before neurosurgical resection. Peritumoral edema was defined as hyperintense signal intensity adjacent to meningiomas on T2-weighted MR images [24]. Tumor area of the meningioma was measured by the two-dimensional diameter method as the sum of the products of both perpendicular diameters of the T1-weighted contrast-enhancing meningioma [25]. Convexity, parasagittal, falx, tentorium, cerebellar convexity, pineal region, and intraventricular meningiomas were defined as non-skull-base meningiomas [26]. Dichotomization of the meningiomas according to location (non-skull-base vs. skull-base) was performed due to known differences regarding histopathology and proliferative potential [21,26]. Laboratory values were recorded using the laboratory information system Lauris (version 17.06.21, Swisslab GmbH, Berlin, Germany). Venous blood samples were routinely analyzed within 24 h prior to surgery for cranial meningiomas. The routine blood examination protocol before surgery as well as the methods of plasma fibrinogen and serum C-reactive protein (CRP) determination were performed as described previously [7].

### 2.3. Histopathology

Neuropathological classification is in line with the 2016 WHO criteria [11]. Immunohistochemical staining was performed in a similar workflow as described before for paraffin-embedded biopsy tissue specimens [27,28]. Determination of the MIB-1 labeling index was performed using the following antibody: anti-Ki67 (Clone Ki-67P, dilution 1:1000, DAKO, Glostrup, Denmark) [22]. Furthermore, semiquantitative assessment and scoring of CD68^+^ staining using anti-CD68 antibodies to determine macrophage infiltrates was performed (Clone KP1, dilution 1:1000, DAKO, Glostrup, Denmark). Meningioma tissues were analyzed for the absence or focal or diffuse staining of CD68^+^ macrophages. Visualization was carried out with diaminobenzidine, and neuropathological investigation was conducted by expert neuropathologists, including A.J.B. Further histopathological analyses were as previously described [29].

### 2.4. Statistical Analysis

Data were recorded and analyzed using SPSS for Windows (version 27.0; IBM Corp, Armonk, NY, USA). Normally distributed data are presented as the mean with the standard deviation (SD). MIB-1 labeling indices as continuous data in patients taking ASA or no ASA were compared and stratified according to known predictors of the MIB-1 labeling index such as sex and location (non-skull-base/skull-base) using the independent *t*-test in order to identify the subgroups benefiting from ASA intake [13,26,30,31]. Further adjustment for age (<60/≥60 years) was performed because patients taking ASA were significantly older. Preoperative demographics, clinical data, imaging characteristics, and inflammatory laboratory markers were compared between female elderly (≥60 years) non-skull-base patients with and without ASA intake using Pearson’s χ2 test (two-sided). Multivariable binary logistic regression analysis was performed to identify predictive variables of a decreased MIB-1 labeling index. The Wald test was used for the analysis of dichotomized variables. A *p*-value of <0.05 was defined as statistically significant. Furthermore, uni- and multivariable analysis of the influence of ASA intake on baseline symptomatic epilepsy in non-skull-base patients was performed.

## 3. Results

### 3.1. Patient Characteristics

A total of 707 patients fulfilled the inclusion criteria and were surgically treated for cranial meningioma at the institutional department. Mean (±SD) age was 61.1 (±13.7) years. The present investigation included 502 females (71.0%) and 205 males (29.0%; female/male ratio 2.5:1). Low-dose ASA intake was found in 107 (15.1%) patients. Mean (±SD) baseline Karnofsky performance scale was 88.7 (±12.2). Tumor classification according to the WHO classification criteria included 556 patients with WHO grade 1 (78.6%) and 151 patients with grade 2 (21.4%). Further characteristics are summarized in Table 1.

### 3.2. Identification of a Subgroup Benefiting from Low-Dose ASA Intake Regarding MIB-1 Labeling Index

Stratification of the cohort by sex, age, and location was performed to identify patients benefiting from low-dose ASA treatment regarding a reduction in the MIB-1 labeling index. The mean (±SD) MIB-1 labeling in the study cohort was 5.4 ± 3.1. Stratification in the sex subgroup revealed that female patients taking ASA had a mean MIB-1 labeling index of 4.7 ± 2.5, whereas female patients without ASA intake had an MIB-1 labeling index of 5.2 ± 2.6 (*p* = 0.16). Female patients were further stratified by age into patients ≥60 years and <60 years in order to analyze the impact of ASA intake on MIB-1 labeling index in those subgroups. Female patients aged 60 years or older taking low-dose ASA on a regular basis had a significantly lower MIB-1 index (4.5 ± 2.3) compared to those in the same age group without ASA intake (5.4 ± 3.0; *p* = 0.045). Further stratification in this identified subgroup was performed by meningioma location (skull-base vs. non-skull-base). Female elderly (≥60 years) non-skull-base meningioma patients taking ASA had a significant lower MIB-1 labeling index (5.0 ± 2.5) compared to those in this subgroup without a regular low-dose ASA medication (6.5 ± 3.3; *p* = 0.02). Figure 1 summarizes the results of this analysis.

### 3.3. Impact of Low-Dose ASA Intake in Female Elderly Non-Skull-Base Meningioma Patients on MIB-1 Index

Univariable analysis of categorical and continuous data among female elderly (≥60 years) non-skull-base meningioma patients with or without ASA intake was performed. Age, physical status reflected by KPS and BMI, regular NSAID treatment for rheumatoid arthritis, regular cortisol intake, serum and plasma inflammatory burden reflected by CRP and fibrinogen levels, tumor area, peritumoral brain edema, edema treatment by dexamethasone intake, Simpson grading, WHO grade, and density of macrophage infiltrates were homogeneously distributed among both groups. Table 2 summarizes the results of the analysis.

Female elderly (≥60 years) non-skull-base meningioma patients with ASA intake had a significant lower MIB-1 labeling index (5.0 ± 2.5) compared to those in this subgroup without a regular low-dose ASA intake (6.5 ± 3.3; *p* = 0.02). Peritumoral brain edema is an imaging characteristic that is known to be strongly associated with the MIB-1 labeling index. Female elderly (≥60 years) non-skull-base meningioma patients with a peritumoral brain edema had a mean MIB-1 labeling index of 6.4 ± 3.8, whereas those without a peritumoral edema had a mean MIB-1 labeling index of 5.9 ± 2.3 (*p* = 0.37). Patients with peritumoral edema are frequently treated with dexamethasone perioperatively, especially when the edema is suspected to be the source of neurological symptoms. Twenty-eight (28/60; 46.7%) patients of the sixty with a peritumoral brain edema received dexamethasone, whereas no patients (0/59; 0.0%) took dexamethasone before surgery in the absence of a peritumoral brain edema (*p* < 0.001). Furthermore, those patients taking dexamethasone for the treatment of a peritumoral brain edema had a significantly lower KPS at 80.4 ± 17.4, and those without dexamethasone treatment had a mean KPS at 88.0 ± 12.4 (*p* = 0.04). Therefore, we performed a multivariable binary logistic regression analysis considering ASA intake, peritumoral brain edema, dexamethasone intake, and Karnofsky performance status (<80/≥80) to identify factors associated with a decreased MIB-1 labeling index (MIB-1 < 5%). The intake of low-dose ASA (adjusted odds ratio (OR): 2.60, 95% confidence interval (CI): 1.03–6.55, *p* = 0.04) was independently and significantly associated with a decreased MIB-1 labeling index (<5%) in female non-skull-base meningioma patients aged 60 years or older. Figure 2 summarizes the results of the multivariable binary logistic regression analysis.

### 3.4. Impact of Low-Dose ASA Intake on Baseline Symptomatic Epilepsy in Non-Skull-Base Meningiomas

Baseline symptomatic epilepsy as a location-specific symptom in non-skull-base meningiomas was found in 24.2% (80/330) of the patients in this anatomic meningioma location. Univariable analysis revealed that age, male sex, ASA intake, and peritumoral brain edema are significantly associated with an increased risk of baseline symptomatic epilepsy in non-skull-base meningiomas. Six (6/80; 7.5%) non-skull-base meningioma patients who took ASA on a regular prescription had baseline symptomatic epilepsy, whereas in the group of patients without baseline symptomatic epilepsy, 22.0% (55/250) of non-skull-base meningioma patients regularly took ASA (*p* = 0.004). Table 3 summarizes the results of the analysis.

The tumor area was not significantly associated with the presence of baseline symptomatic epilepsy. However, tumor area is significantly associated with the presence of a peritumoral brain edema. Patients with a peritumoral brain edema had a mean tumor area of 1641.5 ± 1054.9 mm^2^, and those patients without a peritumoral brain edema had a mean tumor area of 916.7 ± 900.3 mm^2^ (*p* < 0.001). A receiver operating characteristic curve (ROC) was created to determine the optimum cut-off value of tumor area in the prediction of a baseline symptomatic epilepsy (Appendix A). The area under the curve (AUC) for the tumor area is 0.56 (95% CI: 0.48–0.63, *p* = 0.15). The sensitivity and specificity of the preoperative tumor area, with an optimum cut-off set at ≥1186.0 mm^2^, for predicting a symptomatic epilepsy are 60.0% and 54.3%, respectively (Youden’s index: 0.143).

Hence, a multivariable binary logistic regression analysis with consideration of age (<60/≥60 years), sex (female/male), peritumoral brain edema (present/absent), ASA intake (present/absent), and tumor area (<1186.0/≥1186.0) was performed to identify preoperative factors associated with preoperative symptomatic epilepsy in non-skull-base meningiomas. The multivariable analysis found that male sex, age ≥ 60, tumor area ≥1186.0 mm^2^, and the absence of low-dose ASA treatment are significantly associated with preoperative symptomatic epilepsy in non-skull-base meningiomas. Figure 3 displays the results of the multivariable binary logistic regression analysis.

## 4. Discussion

We investigated the impact of regular low-dose ASA intake on proliferative activity in a study cohort of 707 sporadic cranial meningioma patients. Our findings can be summarized into two major points. (1) The intake of low-dose (100 mg) ASA is associated with lower MIB-1 labeling index in female elderly (≥60 years) non-skull-base meningioma patients. (2) Irrespective of sex, COX-2 inhibition by regular low-dose ASA intake is associated with lower rates of symptomatic epilepsy in non-skull-base meningiomas. Therefore, it is very likely that low-dose ASA treatment significantly modifies the inflammatory tumor microenvironment by decreasing the MIB-1 labeling index in female elderly non-skull-base meningioma patients, leading to a reduced risk of symptomatic epilepsy.

### 4.1. COX-2 Inhibition by Low-Dose ASA Intake and MIB-1

We identified that there is a significant difference regarding MIB-1 labeling index in female elderly (≥60 years) non-skull-base meningiomas with or without low-dose ASA treatment. MIB-1 labeling index is an emerging marker in the current era, which strives for maximum tailored treatment scheduling. A recent retrospective study evaluated 239 WHO grade 1 meningioma patients who underwent surgical therapy [32]. In this mentioned series, a recurrence rate of 18.8% was identified for those patients who underwent a gross total resection combined with a MIB-1 labeling index >4.5%. Those characteristics resulted in a similar probability of progression-free survival as in those patients who underwent a subtotal resection. Consequently, the data emphasize the importance of stringently following up with meningioma patients with an increased MIB-1 labeling index even in case of a gross total resection. The association of COX-2 expression and MIB-1 labeling index has been evaluated in a previous clinicopathological study [17]. In this retrospective series, 76 human meningiomas have been investigated regarding the correlation between COX-2 expression and MIB-1 labeling indices as well as WHO grade. The fact that COX-2 expression was significantly associated with increased MIB-1 labeling indices as well as WHO grade support our hypothesis that low-dose ASA intake might decrease the MIB-1 labeling index in a small portion of meningiomas. Furthermore, Ragel et al. [33] revealed that the selective inhibition of COX-2 by celecoxib as an NSAID in meningioma cell lines decreases the MIB-1 labeling index in tumor xenografts. Those important findings were also transferred to a retrospective progression-free-survival analysis stratified by COX-2 expression in 135 completely removed (Simpson grade I-III) intracranial WHO grade 1 and 2 meningiomas [34]. The mentioned series identified that COX-2 expression is an independent risk factor for progression-free survival and was significantly associated with increased VEGF, PDGF, HER2, and MDM2 expression, resulting in increased proliferation. However, there are also studies that could not identify a direct relationship between COX-2 and MIB-1 labeling index in meningiomas [35]. In the series by Lee et al. [35], COX-2 expression in meningiomas was found to be correlated with VEGF level, brain invasion, and necrosis, resulting in increased WHO grades. From a pharmacological point of view, the sex-specific influence of low-dose ASA treatment on the MIB-1 labeling index in cranial non-skull-base meningiomas is somewhat paradoxical so far. However, sex-related differences in drugs with anti-inflammatory properties are highly debated [36]. For instance, throughout the course of life, females have higher numbers of CD4^+^ T cells and CD4/CD8 T-cell ratios [37]. This sex-specific difference of the immune system can also be specified for elderly patients. It could be demonstrated that elderly women have a more active innate and adaptive immune system than men of the same class of age [38]. The sex-specific difference regarding inflammatory microenvironment (e.g., T cell infiltrates, tumor-associated macrophages) might be associated with distinct patterns of chromosome abnormalities and sex chromosomes among male and female meningioma patients. Female meningioma patients have significantly more often an isolated monosomy 22, which has been found to be significantly associated with greater numbers of infiltrating macrophages, natural killer cells, and activated CD69^+^ lymphocytes [39,40]. Furthermore, an immunohistochemical study of 93 meningiomas revealed that female patients have a significantly higher density of fork-head box P3 positive T regulatory cells compared with male patients [41]. T regulatory cells can suppress proliferation, cytokine production, and cytolytic activity of CD4^+^ and CD8^+^ T cells by influencing cell-to-cell contacts and the secretion of cytokines such as TGF-b. Additionally, T regulatory cells can induce an immunosuppressive phenotype in macrophages [42,43]. Moreover, aging influences the inflammatory microenvironment. It was found that there is a significant increase in M2-phenotype macrophage infiltrates with increasing age in 6642 breast cancer patients [44]. With regard to low-dose ASA treatment, ASA causes the formation of anti-inflammatory lipoxins through the acetylation of COX-2 [45]. A randomized clinical trial evaluated whether the administration of ASA results in anti-inflammatory levels of aspirin-triggered 15-epi-lipoxin A_4_ [46]_._ 15-epi-lipoxin A4 is known to have local anti-inflammatory properties in pancreatic and breast cancer, asthma, dermal inflammation, and peritonitis [47,48]. A randomized human trial by Chiang et al. [40] revealed that low-dose ASA treatment has the most pronounced effect regarding the formation of the anti-inflammatory 15-epi-lipoxin A_4_ in elderly women. Hence, the formation of this anti-inflammatory lipid mediator might be an explanation for the sex-specific difference of the low-dose ASA treatment. Furthermore, it was found that 15-epi-lipoxin A_4_ selectively programs the profile of M2-phenotype tumor-associated macrophages and induces a formation of a M1-like profile that triggers tumor cell apoptosis and down-modulates tumor progression in human melanoma [49]. An increased proportion of M2-phenotype macrophages in the tumor microenvironment is also known to have a substantial role in tumor growth and recurrence in meningiomas [50] (Appendix A). However, further research is necessary to evaluate whether low-dose ASA-induced 15-epi-lipoxin A_4_ formation has an anti-inflammatory role in meningioma and whether the sex-specific effect of low-dose ASA regarding 15-epi-lipoxin A_4_ formation also influences the macrophage polarization in human meningiomas. To date, there is no guideline and no substantial evidence supporting the administration of ASA for meningioma patients. Inflammation is a known main avenue in oncogenesis, and there is already evidence supporting the protective benefit of NSAID treatment in specific cancer entities [14,15,16]. As far as NSAID treatment and the risk of central nervous system tumor development is concerned, there is at least evidence from a meta-analysis of observational studies revealing that non-aspirin NSAIDs and ASA use are significantly associated with a lower risk of gliomas but not meningiomas [51]. Furthermore, the results from national registries in Denmark showed a trend that the long-term use (≥5 years) of low-dose ASA slightly reduces the overall risk of glioma compared to patients receiving the medication for less than 5 years [52]. In the field of benign CNS neoplasms, there is especially emerging evidence supporting ASA treatment for vestibular schwannomas to achieve an anti-proliferative effect [53]. Currently, there is an ongoing phase II trial (NCT03079999) evaluating the inhibitory role of aspirin (325 mg twice daily) in the growth of vestibular schwannomas.

### 4.2. Inflammation and Proliferation

As far as the pharmacological modulation of NSAIDs is concerned, tumor-associated macrophages in meningioma have to be brought up because COX-2 is strongly induced in macrophages and further immune cells, which are driven by chronic inflammation [54]. Furthermore, there is a positive feedback loop through prostaglandin E2 and cyclic adenosine monophosphate in the COX-2 expression [55]. Prostaglandin levels have been shown to correlate with the degree of vasogenic edema surrounding supratentorial edemas [56]. Against this backdrop, the potential benefit of NSAID treatment for patients with meningioma might also be caused by an inhibition of the positive feedback loop of COX-2 expression among both meningioma cells and tumor-associated macrophages. Macrophage infiltrates account for approximately 18% of all cells in meningiomas, and their number positively correlates with the WHO grade [57,58]. A retrospective immunohistochemical study of 30 intracranial meningiomas by Proctor et al. [44] revealed that M2 macrophages have a pro-tumoral role, and recurring or WHO grade 2 meningiomas include significantly more M2 macrophages compared to WHO grade 1 meningiomas or primary meningiomas. Conversely, M1 macrophages may act tumoricidal due to the induction of inflammatory reactions via cytotoxic cytokines and leukocyte recruitment, which potentially impede the meningioma growth [59,60,61]. The majority of the macrophage infiltrates are suggested to be polarized to the M2 phenotype. The switch between M1-M2 macrophages is suggested as a key step in the acceleration of cancer aggressiveness [62,63]. COX-2 expression in tumor-associated macrophages was suggested to be essential for the induction and maintenance of the polarization to pro-tumoral M2-macrophages [64]. Furthermore, it was found that COX-2 acts as a key cancer-promoting factor via the induction of a positive feedback loop between macrophages and cancer cells, which might be an avenue for potential therapies in meningioma. In our study, we found no impact of low-dose ASA intake on the number of CD68^+^ macrophage infiltrates. However, it must be reiterated that CD68 staining does not enable a dedicated differentiation between M1 or M2 macrophages.

### 4.3. Symptomatic Epilepsy in Meningiomas and Role of Inflammation

Seizure is a known burden regarding health-related quality of life in meningioma patients [65]. In the present series, 24.2% of all non-skull-base meningioma patients suffered from a symptomatic epilepsy. MIB-1 labeling index is known to be associated with the burden of preoperative epilepsy and the postoperative seizure persistence [66]. Increased MIB-1 labeling indices in meningioma have been demonstrated to be strongly correlated with COX-2 expression [17,18]. COX-2 has been suggested to be upregulated in various cells within the CNS following seizures, which results in the increased production of inflammatory mediators and prostaglandins aggravating the frequency and severity of seizure [67]. Hence, it is proposed that COX-2 inhibition may reduce the seizure burden as well as the pharmacoresistance to antiepileptic drugs [19]. Several pathophysiological mechanisms of COX-2 in symptomatic epilepsy are highly debated (e.g., anti-inflammatory, blood–brain barrier disruption). COX-2 inhibition is suggested to aid as an adjunctive to antiepileptic drug therapy due to the potential to prevent an upregulation of the multidrug transporter P-glycoprotein, which is often overexpressed in drug-resistant epilepsy [68]. Regarding non-selective COX-2 inhibitors such as ASA, it has been identified that ASA can independently or as an adjunct to antiepileptic drug treatment improve the burden of seizure in epilepsy and phacomatoses. The majority of studies supporting the benefit of ASA treatment regarding epilepsy treatment were performed on Sturge–Weber syndrome patients. Lance et al. [69] evaluated the effect of low-dose (3–5 mg/kg/d) ASA treatment in 58 Sturge–Weber syndrome patients with brain involvement and achieved seizure control in 91% of the patients. The beneficial effects of ASA regarding epilepsy are not only described for patients with Sturge–Weber syndrome, but there are also the data of Godfred et al. [70], who retrospectively analyzed whether ASA alters the seizure frequency in adults with focal onset epilepsy while undergoing video-EEG monitoring. They found that patients with focal epilepsy taking ASA have significantly fewer seizures compared to age-, sex-, and disease-matched patients without ASA intake.

### 4.4. Limitations

Generally, the retrospective nature and the corresponding risk of incomplete documentation is the major limitation of the present investigation. Nevertheless, the regular low-dose ASA treatment was well-recorded because it is of paramount importance to discontinue anti-platelets before intracranial surgery if it is deemed possible. Low-dose ASA treatment was indicated in all cases due to the presence of cardiovascular diseases or risk factors for cardiovascular events. Hence, in all patients, ASA intake was prescribed with a low-dose regimen (100 mg once daily) for months to several years. Furthermore, we cannot provide accurate data regarding the duration of low-dose ASA treatment in each patient, which might inform future trials from a pharmacological point of view. However, patients who sporadically took ASA for pain relief (e.g., headache) were not included. The second important limitation in our retrospective study is the lack of immunohistochemical staining of COX2, which might reveal a more profound insight into which subgroup of patients might benefit most from future anti-inflammatory treatments via COX-2 inhibition.

## 5. Conclusions

Low-dose ASA treatment significantly decreases the MIB-1 labeling index in female elderly non-skull-base meningioma patients. Moreover, regular use of low-dose ASA might reduce the risk of symptomatic epilepsy in non-skull-base meningiomas.

## Figures and Tables

**Figure 1 cancers-14-04285-f001:**
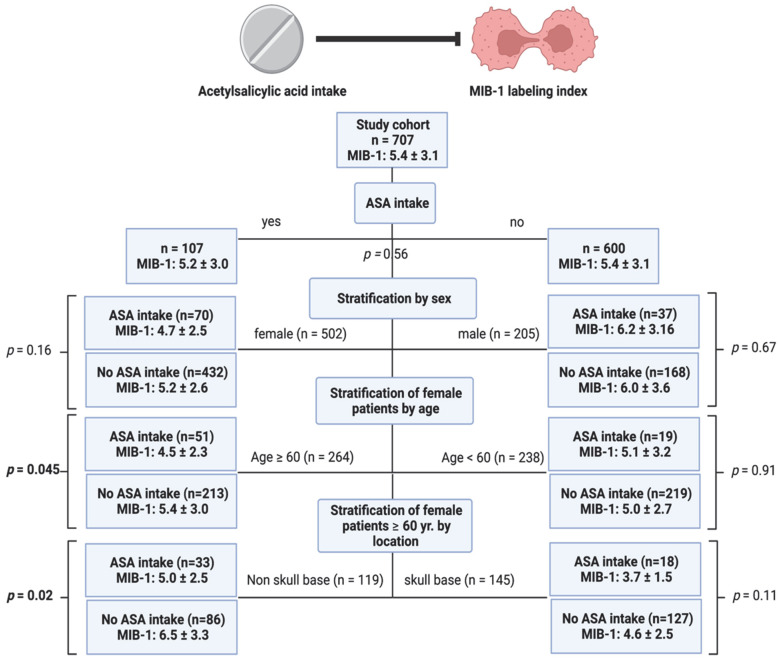
Flow diagram displaying the stratification of patients with or without ASA intake in the subgroups of sex and further stratification of female patients by age and elderly females by meningioma location. Statistical results of the independent t-test comparing the mean values and the corresponding standard deviations of the MIB-1 labeling indices are given. Significant statistical results are presented in bold.

**Figure 2 cancers-14-04285-f002:**
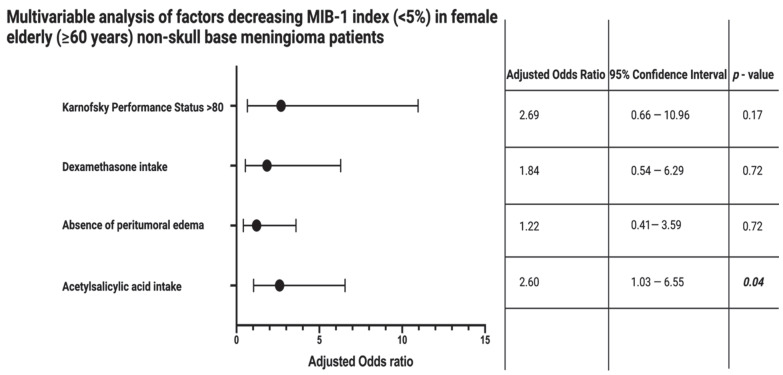
Forest plots from multivariable binary logistic regression analysis: Acetylsalicylic acid intake is an independent factor associated with an MIB-1 labeling index <5% in female elderly (≥60 years) non-skull-base meningioma patients. *p*-values in italics and bold display statistically significant results.

**Figure 3 cancers-14-04285-f003:**
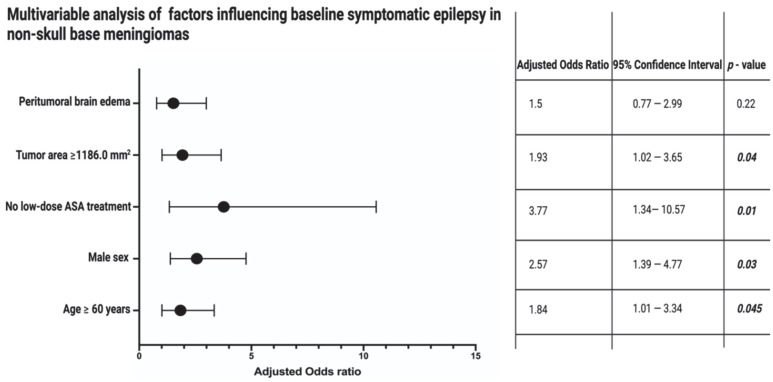
Forest plots from multivariable binary logistic regression analysis: Absence of low-dose ASA intake, male sex, age ≥ 60 years, and tumor area ≥1186.0 mm^2^ are independent predictors of a preoperative symptomatic epilepsy in non-skull-base meningiomas. *p*-values in bold and italics display statistically significant results.

**Table 1 cancers-14-04285-t001:** Patient characteristics of the study cohort (*n* = 707).

Characteristics	N (%)
Sex	
Female	502 (71.0%)
Male	205 (29.0%)
Age, mean ± SD	61.1 ± 13.7
Body mass index, mean ± SD	27.3 ± 5.9
ASA intake	107 (15.1%)
Preoperative KPS, mean ± SD	88.7 ± 12.2
Plasma fibrinogen, mean ± SD	3.0 ± 0.9
Serum c-reactive protein, mean ± SD	3.7 ± 7.5
Peritumoral brain edema	303 (42.9%)
Location	
Non-skull-base meningioma	327 (46.3%)
Skull-base meningioma	380 (53.7%)
WHO grade	
1	556 (78.6%)
2	151 (21.4%)
MIB-1 labeling index, mean ± SD	5.4 ± 3.1
Mitotic figures, mean ± SD	1.9 ± 2.6

KPS = Karnofsky performance status; MIB-1 = molecular immunology borstel-1; SD = standard deviation.

**Table 2 cancers-14-04285-t002:** Baseline clinical, laboratory, and imaging characteristics of female elderly (≥60 years) non-skull-base meningioma patients with or without acetylsalicylic acid intake.

Characteristics	No ASA Intake% (86/119; 72.3%)	ASA Intake (33/119; 27.7%)	*p*-Value
Age (years), mean ± SD	72.0 ± 77	71.2 ± 8.4	0.64
Preoperative KPS, mean ± SD	86.9 ± 13.9	84.1 ± 14.6	0.33
Body mass index, mean ± SD	26.7 ± 5.5	27.8 ± 5.4	0.35
Preoperative KPS, mean ± SD	86.2 ± 10.4	84.7 ± 9.6	0.70
Plasma fibrinogen, mean ± SD	3.0 ± 1.0	3.3 ± 1.0	0.19
Rheumatoid arthritis with NSAID treatment	2 (2.3%)	1 (3.0%)	0.99
Regular cortisol intake	14 (16.3%)	7 (21.2%)	0.59
Serum c-reactive protein, mean ± SD	4.9 ± 10.1	4.7 ± 6.8	0.90
Tumor area, mean ± SD, mm^2^	1476.5 ± 988.9	1435.6 ± 974.8	0.85
Peritumoral brain edema			0.84
Present	44 (51.2%)	16 (48.5%)	
Not present	42 (48.8%)	17 (51.5%)	
Preoperative dexamethasone intake			0.81
Present	21 (24.4%)	7 (21.2%)	
Absent	65 (75.6%)	26 (78.8%)	
Simpson grade			0.99
≤III	82 (95.3%)	32 (97.0%)	
>III	4 (4.7%)	1 (3.0%)	
WHO grade			0.07
1	58 (67.4%)	28 (84.8%)	
2	28 (32.6%)	5 (15.2%)	
CD68^+^ macrophages (available in 91 patients)			0.61
Diffuse	47 (73.4%)	18 (66.7%)	
Focal	17 (26.6%)	9 (33.3%)	

KPS = Karnofsky performance status; MIB-1 = molecular immunology borstel-1; SD = standard deviation.

**Table 3 cancers-14-04285-t003:** Baseline clinical, laboratory, and imaging characteristics of non-skull-base meningioma patients with or without preoperative symptomatic epilepsy.

Characteristics	No Seizure (250/330; 75.8%)	Seizure (80/330; 24.2%)	*p*-Value
Age (years), mean ± SD	57.6 ± 14.6	62.9 ± 13.9	0.004
Sex			0.01
Female	178/223 (79.8%)	45/223 (20.2%)	
Male	72/107 (67.3%)	35/107 (32.7%)	
Preoperative KPS, mean ± SD	89.4 ± 12.5	89.7 ± 11.7	0.82
Body mass index, mean ± SD	27.4 ± 5.5	28.0 ± 6.5	0.44
ASA intake	55 (22.0%)	6 (7.5%)	0.004
Tumor area, mean ± SD, mm^2^	1379.9 ± 1034.8	1601.1 ± 1233.4	0.13
Peritumoral brain edema			0.0001
Present	99 (39.6%)	55 (68.8%)	

ASA = acetylsalicylic acid; KPS = Karnofsky performance status; SD = standard deviation.

## Data Availability

All data are included in this manuscript.

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
