# Peer review of "Low-Dose Acetylsalicylic Acid Treatment in Non-Skull-Base Meningiomas: Impact on Tumor Proliferation and Seizure Burden"

_cancers, 2022, doi:10.3390/cancers14174285_

Round 1
Reviewer 1 Report
This paper indicate that acetylsalicylic acid (ASA) treatment decreased MIB-1 and preoperative seizure burden in female non-skull base meningioma patients,providing important clinic experience for future meningioma therapy. I think this paper can be accepted for publication after minor language modification.
The main question addressed by the research:
Acetylsalicylic acid is a kind of clinical approval drug and if it really can benefit female non-skull base meningiomas patients as authors described in their clinic therapy. This study may provide direct help for those meningiomas patients.
original or relevant of the topic:
I think it is relevant, clinicians keep trying different kinds of drugs on patients, there are thousands of. papers about this, but it is very hard to identify a certain drug that can benefit patient in clinic, especially cancer patients .
Its add to the subject area compared to other published material:
As I mention previously, it is very hard to find a drug that can benefit patient in clinic. Other papers may try different kinds of drugs to inhibit cancer, but some studies only have in vitro experiment, and others may use trial drug in patient. However, in this study, acetylsalicylic acid is a kind of clinical approval drug.
Regarding the methodology, what specific improvements could the authors consider:
I think they provide enough clinical indexes to support their conclusion. Maybe author can do a combination treatment, use acetylsalicylic acid combine with some tradition anti-cancer drugs for meningiomas. Consistency of conclusions with the evidence and arguments presented:
Actually, the idea of paper is simple, they just try to use acetylsalicylic acid to treat meningiomas patients. The clinic indexes provide enough evidence that acetylsalicylic acid indeed alleviate the diseases. But about the mechanism, they explain nothing. The additional comments on the tables and figures:
I have no comments on the tables and figures. However, this study lack mechanism study, could authors just at least try to explain the mechanism how acetylsalicylic acid benefit patients, why only female patients, as it is not a tradition anti-cancer drug.
Author Response
Dear Reviewer
Thank you for reading our manuscript and critically reviewing it, which will help us improve it to a better scientific level and make it more understandable to the readership.
We agree with the reviewer that the sex-specific influence of low-dose ASA in elderly female meningioma patients is somewhat paradoxical so far. Hence, we have revised the suggested hypothesis explaining the pharmacological modulation by low-dose ASA treatment in the section “4.1 COX-2 inhibition by low-dose ASA intake and MIB-1”. Furthermore, we newly created a supplementary figure S2, which illustrates our suggested hypothesis regarding the specific impact of low-dose ASA treatment on MIB-1 labeling index in elderly female non-skull base meningioma patients. As far as low-dose ASA treatment is concerned, ASA induces the formation of anti-inflammatory lipoxins through the acetylation of COX-2 [1]. A prospective randomized clinical study investigated whether the administration of ASA results in anti-inflammatory levels of aspirin-triggered 15-epi-lipoxin A4 [2]. 15-epi-lipoxin A4 has local anti-inflammatory properties in other diseases such as pancreatic, breast cancer, asthma, dermal inflammation, or peritonitis [3, 4]. This randomized human trial by Chiang et al. [2] found that low-dose ASA treatment has the most pronounced effect regarding the formation of the anti-inflammatory 15-epi-lipoxin A4 in elderly women. Hence, the formation of this anti-inflammatory lipid mediator might be an explanation for the sex-specific difference of the low-dose ASA treatment in human meningiomas. Furthermore, it was found that 15-epi-lipoxin A4 selectively influences the profile of M2-phenotype tumor-associated macrophages and induces a formation of a M1-like profile which triggers tumor cell apoptosis and attenuates tumor progression in human melanoma [5]. M2-phenotype macrophages in the tumor microenvironment are also known to have a substantial role in tumor growth and recurrence in meningiomas [6]. However, further research is necessary to investigate whether low-dose ASA induced 15-epi-lipoxin A4 formation has anti-inflammatory properties in meningioma and whether the sex-specific effect of low-dose ASA regarding 15-epi-lipoxin A4 formation also influences the macrophage polarization in human meningiomas.
Furthermore, the reviewer is absolutely right that ASA is not a traditional anti-cancer drug and ASA might be also interesting in terms of a combined therapy with an anti-cancer drug. We strive to investigate the use of COX-2 inhibition in a prospective phase II a study of human cranial meningiomas which might also give us more insight regarding the influence on immune cell infiltrates, macrophages polarization, and proliferative activity. Furthermore, the manuscript underwent a language editing by a native speaker.
References
- Chiang, N.; Bermudez, E.A.; Ridker, P.M.; Hurwitz, S.; Serhan, C.N. Aspirin triggers anti-inflammatory 15-epi-lipoxin A4 and inhibits thromboxane in a randomized human trial. Natl. Acad. Sci. USA. 2004, 101, 15178-15183
- Chiang, N.; Hurwitz, S.; Ridker, P.M.; Serhan, C.N. Aspirin has a gender-dependent impact on anti-inflammatory 15-epi-lipoxin A4 formation: A randomized human trial. Thromb. Vasc. Biol. 2006, 26, e14-e17
- Chiang, N.; Arita, M.; Serhan, C.N. Anti-inflammatory circuitry: lipoxin, aspirin-triggered lipoxins and their receptor ALX. Prostaglandins Leukot Essent Fatty Acids. 2005, 73, 163-177
- Schnittert, J.; Heinrich, M.A.; Kuninty, P.R.; Storm, G.; Prakash, J. Reprogramming tumor stroma using an endogenous lipid lipoxin A4 to treat pancreatic cancer. Cancer Lett. 2018, 420, 247-258
- Simões, R.L.; De-Brito, N.M.; Cunha-Costa, H.; Morandi, V.; Fierro, I.M.; Roitt, I.M.; Barja-Fidalgo, C. Lipoxin A4selectively programs the profile of M2 tumor-associated macrophages which favour control of tumor progression. Int J Cancer. 2017, 140(2), 346-357
- Proctor, D.T.; Huang, J.; Lama, S.; Albakr, A.; Van Marle, G.; Sutherland, G.R. Tumor-associated macrophage infiltration in meningioma. Neurooncol Adv. 2019, 1(1), vdz018.
Reviewer 2 Report
This is a manuscript on an interesting and relevant topic. However, the manuscript needs some additional work. In addition it is not well written and needs review by a native speaker
Introduction:
Meningioma arise from achanoid cap and dural border cells.
The reviewer is not aware of a chemotherapy for meningiomas, which has proven to be effective.
Can the authors specific what they mean by high class evidence.
Results:
What is rational for the stratification into non-skull based meningioma, especially the grouping used in this manuscript?
The fact that over 60's show a significant effect from ASS leads to question if that effect was related duration of use of ASS, please comment.
After multivariant analysis statistical significance is see only in a subgroup female over 60 non skull base. In the discussion the authors somehow try to explain that. That explanation is a bit weak, what one what would really like to see is data or at least good references of T-cells in meningioma of elderly female vs others.
Discussion:
Too long and sometimes drifting off into unrelated areas
Author Response
Dear Reviewer
Thank you for reading our manuscript and critically reviewing it, which will help us improve it to a better scientific level and make it more understandable to the readership.
In the following we would like to respond to your remarks:
The reviewer is absolutely right that meningiomas derive from two types of cells related to arachnoid villi: arachnoid cells and dural border cells [1]. Furthermore, we have specified the meaning of the term “high-class evidence”. To date, there is no prospective data elucidating the influence or potential benefit of COX-2 inhibition in human meningiomas. Hence, there is the need for reliable data from a prospective randomized phase trial supporting the clinical benefit or at least the proof of concept of COX-2 inhibition in human cranial meningiomas.
Cranial meningiomas were stratified into “skull base” and “non-skull base” meningiomas because of known differences regarding histopathology and proliferative activity. Maiuri et al. [2] retrospectively reviewed 300 human meningiomas regarding WHO grades, MIB-1 labeling indices, progesterone receptor expression and histopathological subtype. It was found that the frequency of WHO grade 2 meningiomas and the rates of increased MIB-1 labeling indices are significantly higher among non-skull base meningiomas compared to skull-base meningiomas and spinal ones [2]. In a recent institutional series, we could also confirm that non-skull base meningiomas have increased MIB-1 labeling indices compared to skull base meningiomas [3]. Therefore, we have added this information to the methods and revised the section “2.2 Data Recording and radiological features” according to the suggestion by the reviewer in order to enhance the understanding of the manuscript.
We absolutely agree with the reviewer that the duration of low-dose ASA treatment might play an important role from a pharmacological point of view. Hence, we have a heterogeneous collective regarding the duration of low-dose ASA treatment and not all patients received low-dose ASA for the same duration. However, patients who sporadically took ASA for pain relief (e.g., headache) were strictly excluded. Unfortunately, the present investigation is limited by the retrospective nature, and we are not able to provide accurate information regarding the duration of ASA treatment for each patient. Nevertheless, we strive to prospectively investigate the role of COX-2 inhibition in human meningioma patients who will receive the drug treatment at the same timepoint. We added this important remark and limitation of our investigation to the section “4.4 Limitations”.
We agree with the reviewer that the sex-specific influence of low-dose ASA in elderly female meningioma patients is somewhat paradoxical so far. Hence, we have revised the suggested hypothesis explaining the pharmacological modulation by low-dose ASA treatment in the section “4.1 COX-2 inhibition by low-dose ASA intake and MIB-1”. Furthermore, we newly created a supplementary figure S2, which illustrates our suggested hypothesis regarding the specific impact of low-dose ASA treatment on MIB-1 labeling index in elderly female non-skull base meningioma patients. As far as low-dose ASA treatment is concerned, ASA induces the formation of anti-inflammatory lipoxins through the acetylation of COX-2 [4]. A prospective randomized clinical study investigated whether the administration of ASA results in anti-inflammatory levels of aspirin-triggered 15-epi-lipoxin A4 [5]. 15-epi-lipoxin A4 has local anti-inflammatory properties in other diseases such as pancreatic, breast cancer, asthma, dermal inflammation, or peritonitis [6, 7]. This randomized human trial by Chiang et al. [5] found that low-dose ASA treatment has the most pronounced effect regarding the formation of the anti-inflammatory 15-epi-lipoxin A4 in elderly women. Hence, the formation of this anti-inflammatory lipid mediator might be an explanation for the sex-specific difference of the low-dose ASA treatment in human meningiomas. Furthermore, it was found that 15-epi-lipoxin A4 selectively influences the profile of M2-phenotype tumor-associated macrophages and induces a formation of a M1-like profile which triggers tumor cell apoptosis and attenuates tumor progression in human melanoma [8]. M2-phenotype macrophages in the tumor microenvironment are also known to have a substantial role in tumor growth and recurrence in meningiomas [9]. However, further research is necessary to investigate whether low-dose ASA induced 15-epi-lipoxin A4 formation has anti-inflammatory properties in meningioma and whether the sex-specific effect of low-dose ASA regarding 15-epi-lipoxin A4 formation also influences the macrophage polarization in human meningiomas. The reviewer is absolutely right that t-cells play a crucial role in the tumor immune microenvironment. Tumor-infiltrating lymphocytes in meningiomas are predominantly T-cells. In anaplastic meningiomas (WHO grade 3) the proportion of infiltrating regulatory T-cells is significantly increased, whereas the number of CD4 and CD8 T-cells is low in those high-grade meningiomas [10]. The majority of T-cell infiltrates in high-grade meningiomas consist of exhausted PD1+ T cells and regulatory T-cells [11, 12]. This proportion has also clinical implications for the patients because an increased proportion of PD1+ T cell infiltrates results in a poorer probability of progression of free survival [12]. Unfortunately, we cannot provide specific own data regarding immunohistochemical t-cell labeling which might give more insight into this potential modulation of the inflammatory microenvironment by low-dose ASA intake. Unfortunately, we cannot provide specific data regarding immunohistochemical t-cell labeling or macrophage polarization which might give more insight into this potential modulation of the inflammatory microenvironment by low-dose ASA intake. However, we strive to investigate the influence of COX-2 inhibition in human cranial meningiomas in a prospective phase IIa trial, which will also provide more information regarding the modulation of the macrophage polarization and density of infiltrating t-cells. The other sections of the discussion were shortened to strengthen the focus of the manuscript on the influence of low-dose ASA treatment on MIB-1 labeling indices reduction in female non-skull base meningioma patients and the reduction of the seizure burden. Furthermore, the manuscript underwent a language editing by a native speaker.
References
- Hosainey, S.A.M.; Bouget, D.; Reinertsen, I.; Sagberg, L.M.; Torp, S.H.; Jakola, A.S.; Solheim, O. Are there predilection sites for intracranial meningioma? A population-based atlas. Neurosurg Rev 2022, 45(2), 1543-1552
- Maiuri, F.; Mariniello, G.; Guadagno, E.; Barbato, M.; Corvino, S.; Del Basso De Caro, M. WHO grade, proliferation index, and progesterone receptor expression are different according to the location of meningioma. Acta Neurochir. (Wien) 2019, 161, 2553–2561
- Wach, J.; Lampmann, T.; Güresir, Á.; Vatter, H.; Herrlinger, U.; Becker, A.; Cases-Cunillera, S.; Hölzel, M.; Toma, M.; Güresir, E. Proliferative Potential, and Inflammatory Tumor Microenvironment in Meningioma Correlate with Neurological Function at Presentation and Anatomical Location-From Convexity to Skull Base and Spine. Cancers (Basel) 2022, 14
- Chiang, N.; Bermudez, E.A.; Ridker, P.M.; Hurwitz, S.; Serhan, C.N. Aspirin triggers anti-inflammatory 15-epi-lipoxin A4 and inhibits thromboxane in a randomized human trial. Natl. Acad. Sci. USA. 2004, 101, 15178-15183
- Chiang, N.; Hurwitz, S.; Ridker, P.M.; Serhan, C.N. Aspirin has a gender-dependent impact on anti-inflammatory 15-epi-lipoxin A4 formation: A randomized human trial. Thromb. Vasc. Biol. 2006, 26, e14-e17
- Chiang, N.; Arita, M.; Serhan, C.N. Anti-inflammatory circuitry: lipoxin, aspirin-triggered lipoxins and their receptor ALX. Prostaglandins Leukot Essent Fatty Acids. 2005, 73, 163-177
- Schnittert, J.; Heinrich, M.A.; Kuninty, P.R.; Storm, G.; Prakash, J. Reprogramming tumor stroma using an endogenous lipid lipoxin A4 to treat pancreatic cancer. Cancer Lett. 2018, 420, 247-258
- Simões, R.L.; De-Brito, N.M.; Cunha-Costa, H.; Morandi, V.; Fierro, I.M.; Roitt, I.M.; Barja-Fidalgo, C. Lipoxin A4selectively programs the profile of M2 tumor-associated macrophages which favour control of tumor progression. Int J Cancer. 2017, 140(2), 346-357
- Proctor, D.T.; Huang, J.; Lama, S.; Albakr, A.; Van Marle, G.; Sutherland, G.R. Tumor-associated macrophage infiltration in meningioma. Neurooncol Adv. 2019, 1(1), vdz018.
- Du, Z.; Abedalthagafi, M.; Aizer, A.A.; Mchenry, A.R.; Sun, H.H.; Bray, M.A.; et al. Increased expression of the immune modulatory molecule PD-L1 (CD274) in anaplastic meningioma. Oncotarget. 2015, 6, 4704-4716
- Fang, L.; Lowther, D.E.; Meizlish, M.L.; Anderson, R.C.; Bruce, J.N.; Devine, L.; et al. The immune cell infiltrate populating meningiomas is composed of mature, antigen-experienced T and B cells. Neuro Oncol. 2013, 15, 1479-1490
- Li, Y.D.; Veliceasa, D.; Lamano, J.B.; Lamano, J.B.; Kaur, G.; Biyashev, D.; et al. Systemic and local immunosuppression in patients with high-grade meningiomas. Cancer Immunol Immunother. 2019, 68, 999-1009
Reviewer 3 Report
I read with much interest the article entitled “Low-dose acetylsalicylic acid treatment in non-skull base meningiomas: Impact on tumor proliferation and seizure burden” by Johannes Wach et al. I think that the Authors described a very interesting topic. However, the manuscript shows some weakness that can be improved. Although the Authors conducted a correct statistical analyses, some considerations have to be clarified. In fact, the association between the inflammatory tumoral microenvironment and the influence of ASA is not clear. It probably needs clarification from a strictly biochemical or pathological point of view, having been evaluated only in statistical terms. This is an important aspect to be evaluated as the authors report that it would lead to a reduction in epileptic seizures. Also, the relationships with appearance of macrophage infiltration needs some improvement. The correlations of these infiltrates (macrophages / microglia) with their both the protumoral and antitumoral effect should be better examined. Some other interesting work published in the literature has not been cited and considered.
1. Association between nonsteroidal anti-inflammatory drugs use and risk of central nervous system tumors: a dose-response meta analysis, Tao Zhang , Xiaowen Yang Oncotarget. 2017 Oct 11;8(60):102486-102498. doi: 10.18632/oncotarget.21829.
2. Use of low-dose aspirin and non-aspirin nonsteroidal anti-inflammatory drugs and risk of glioma: a case-control study. Gaist D, García-Rodríguez LA, Sørensen HT, Hallas J, Friis S. Br J Cancer. 2013 Mar 19;108(5):1189-94. doi: 10.1038/bjc.2013.87
I think that in this current form it is not suitable for publication.
Author Response
Dear Reviewer
Thank you for reading our manuscript and critically reviewing it, which will help us improve it to a better scientific level and make it more understandable to the readership.
In the following we would like to respond to your remarks:
We agree with the reviewer that the biochemical or neuropathological aspect regarding low-dose ASA treatment in non-skull base meningiomas is still unclear. Furthermore, the sex-specific influence of low-dose ASA in elderly female meningioma patients is also somewhat paradoxical so far. Hence, we have revised the suggested hypothesis explaining the pharmacological modulation by low-dose ASA treatment in the section “4.1 COX-2 inhibition by low-dose ASA intake and MIB-1”. Furthermore, we newly created a supplementary figure S2, which illustrates our suggested hypothesis regarding the specific impact of low-dose ASA treatment on MIB-1 labeling index in elderly female non-skull base meningioma patients. As far as low-dose ASA treatment is concerned, ASA induces the formation of anti-inflammatory lipoxins through the acetylation of COX-2 [1]. A prospective randomized clinical study investigated whether the administration of ASA results in anti-inflammatory levels of aspirin-triggered 15-epi-lipoxin A4 [2]. 15-epi-lipoxin A4 has local anti-inflammatory properties in other diseases such as pancreatic, breast cancer, asthma, dermal inflammation, or peritonitis [3, 4]. This randomized human trial by Chiang et al. [2] found that low-dose ASA treatment has the most pronounced effect regarding the formation of the anti-inflammatory 15-epi-lipoxin A4 in elderly women. Hence, the formation of this anti-inflammatory lipid mediator might be an explanation for the sex-specific difference of the low-dose ASA treatment in human meningiomas. Furthermore, it was found that 15-epi-lipoxin A4 selectively influences the profile of M2-phenotype tumor-associated macrophages and induces a formation of a M1-like profile which triggers tumor cell apoptosis and attenuates tumor progression in human melanoma [5]. M2-phenotype macrophages in the tumor microenvironment are also known to have a substantial role in tumor growth and recurrence in meningiomas [6]. However, further research is necessary to investigate whether low-dose ASA induced 15-epi-lipoxin A4 formation has anti-inflammatory properties in meningioma and whether the sex-specific effect of low-dose ASA regarding 15-epi-lipoxin A4 formation also influences the macrophage polarization in human meningiomas. Unfortunately, we cannot provide specific data regarding immunohistochemical t-cell labeling or macrophage polarization which might give more insight into this potential modulation of the inflammatory microenvironment by low-dose ASA intake. However, we strive to investigate the influence of COX-2 inhibition in human cranial meningiomas in a prospective trial, which will also provide more information regarding the modulation of the macrophage polarization and density of infiltrating t-cells. Furthermore, the reviewer is absolutely right that ASA is not a traditional anti-cancer drug and ASA might be also interesting in terms of a combined therapy with an anti-cancer drug. We strive to investigate the use of COX-2 inhibition in a prospective phase II a study of human cranial meningiomas which might also give us more insight regarding the influence on immune cell infiltrates, macrophages polarization, and proliferative activity.
Moreover, we have integrated the suggested references regarding the evidence of NSAID, non-aspirin NSAID, and ASA treatment in the risk of central nervous system (CNS) tumors. As far as NSAID treatment and risk of CNS tumors development is concerned, there are interesting data from a meta-analysis investigating observational studies, which revealed that non-aspirin NSAIDs and ASA use are significantly associated with a lower risk of gliomas, but not meningiomas [7]. Additionally, analyses of national registries from Denmark showed that there is a trend toward the benefit of long-term (≥5 years) low-dose ASA treatment compared to patients receiving the medication less than 5 years regarding the overall risk of glioma [8]. This revised part was also included in the section “4.1 COX-2 inhibition by low-dose ASA intake and MIB-1”.
References
- Chiang, N.; Bermudez, E.A.; Ridker, P.M.; Hurwitz, S.; Serhan, C.N. Aspirin triggers anti-inflammatory 15-epi-lipoxin A4 and inhibits thromboxane in a randomized human trial. Natl. Acad. Sci. USA. 2004, 101, 15178-15183
- Chiang, N.; Hurwitz, S.; Ridker, P.M.; Serhan, C.N. Aspirin has a gender-dependent impact on anti-inflammatory 15-epi-lipoxin A4 formation: A randomized human trial. Thromb. Vasc. Biol. 2006, 26, e14-e17
- Chiang, N.; Arita, M.; Serhan, C.N. Anti-inflammatory circuitry: lipoxin, aspirin-triggered lipoxins and their receptor ALX. Prostaglandins Leukot Essent Fatty Acids. 2005, 73, 163-177
- Schnittert, J.; Heinrich, M.A.; Kuninty, P.R.; Storm, G.; Prakash, J. Reprogramming tumor stroma using an endogenous lipid lipoxin A4 to treat pancreatic cancer. Cancer Lett. 2018, 420, 247-258
- Simões, R.L.; De-Brito, N.M.; Cunha-Costa, H.; Morandi, V.; Fierro, I.M.; Roitt, I.M.; Barja-Fidalgo, C. Lipoxin A4selectively programs the profile of M2 tumor-associated macrophages which favour control of tumor progression. Int J Cancer. 2017, 140(2), 346-357
- Proctor, D.T.; Huang, J.; Lama, S.; Albakr, A.; Van Marle, G.; Sutherland, G.R. Tumor-associated macrophage infiltration in meningioma. Neurooncol Adv. 2019, 1(1), vdz018.
- Zhang, T.; Yang, X.; Liu, P.; Zhou, J.; Luo, J.; Wang, H.; Li, A.; Zhou, Y. Association between nonsteroidal anti-inflammatory drugs use and risk of central nervous system tumors: a dose-response meta analysis. Oncotarget. 2017, 8(60), 102486-102498.
- Gaist, D.; García-Rodríguez, L.A.; Sørensen, H.T.; Hallas, J.; Friis, S. Use of low-dose aspirin and non-aspirin nonsteroidal anti-inflammatory drugs and risk of glioma: a case-control study. Br J Cancer. 2013, 108(5), 1189-94.
Round 2
Reviewer 2 Report
Somewhat improved.
Still would benefit from showing or citing age and gender difference in T cell or macrophage infiltration in meningioma
Author Response
Dear Reviewer
Thank you for reading our manuscript and critically reviewing it, which will help us improve it to a better scientific level and make it more understandable to the readership.
We agree with the reviewer that the sex-specific influence on inflammatory infiltrates is important. This potential sex-specific difference might be also associated with distinct patterns of chromosome abnormalities and sex chromosomes among male and female meningioma patients. Meningiomas with an isolated monosomy 22 were found to have significantly greater numbers of infiltrating tissue macrophages, natural killer cells, and activated CD69+ lymphocytes compared to meningiomas with diploid and complex karyotypes. Those findings are based on the investigations of Domingues et al. [1], which analyzed 75 meningioma patients by multiparameter flow cytometry, clinic-biological correlations, cytogenetic and gene expression profile evaluation. Furthermore, the same group performed an interphase fluorescence in situ hybridization (iFISH) in 164 patients [2]. The iFISH investigations revealed a higher frequency of isolated monosomy 22 in female patients. Hence, different patterns of chromosome abnormalities and gene-expression profiles are associated with the patient sex and influences the inflammatory microenvironment. Furthermore, Ding et al. [3] performed an immunohistochemical analysis (CD4, CD20, CD68, FOXP3) of the subtypes of cells in the tumor microenvironment and correlated the parameters with clinical characteristics in 93 meningioma patients. They found that female patients have a higher density of fork-head box P3 (Foxp3+) T regulatory cells (Tregs) compared with male meningioma patients. Tregs have the capability to attenuate proliferation, cytokine production, and cytolytic activity of CD4+ and CD8+ T cells by influencing cell-to-cell contacts and the secretion of cytokines such as TGF-b. Moreover, Tregs can induce an immunosuppressive phenotype in macrophages [4, 5].
We agree with the reviewer regarding the potential impact of age on the immune landscape of meningioma tissue. There are strong data showing a significant increase of M2 macrophage infiltrates with increasing age in 6,642 breast cancer patients [6]. To date, we do not know the impact of aging on the immune microenvironment (e.g., macrophage polarization-shift in elderly meningioma patients) in meningiomas. Hence, the raised remark of the reviewer is of paramount importance for further studies investigating both immune microenvironment and anti-inflammatory therapies in meningiomas. We strive to investigate this issue as a secondary endpoint in a prospective randomized phase IIa trial.
References
- Domingues, P.H.; Teodosio, C.; Otero, A.; Sousa, P.; Ortiz, J.; Macias Mdel, C.; Goncalves, J.M.; Nieto, A.B.; Lopes, M.C.; de Oliveira, C.; Orfao, A.; Tabernero, M.D. Association between inflammatory infiltrates and isolated monosomy 22/del(22q) in meningiomas. PLoS One. 2013, 8(10), e74798
- Tabernero, M.D.; Espinosa, A.B.; Maillo, A.; Rebelo, O.; Vera, J.F.; Sayagues, J.M.; Merino, M.; Diaz, P.; Sousa, P.; Orfao, A. Patient gender is associated with distinct patterns of chromosomal abnormalities and sex chromosome linked gene-expression profiles in meningiomas. Oncologist. 2007, 12(10), 1225-35
- Ding, Y.; Qiu, L.; Xu, Q.; Song, L.; Yang, S.; Yang, T. Relationships between tumor microenvironment and clinicopathological parameters in meningioma. Int J Clin Exp Pathol. 2014, 7(10), 6973-9
- Kryczek, I.; Wei, S.; Zhu, G.; Myers, L.; Mottram, P.; Cheng, P.; Chen, L.; Coukos, G.; Zou, W. Relationship between B7-H4, regulatory T cells, and patient outcome in human ovarian carcinoma. Cancer Res. 2007, 67(18), 8900-5
- Milne, K.; Köbel, M.; Kalloger, S.E.; Barnes, R.O.; Gao, D.; Gilks, C.B.; Watson, P.H.; Nelson, B.H. Systematic analysis of immune infiltrates in high-grade serous ovarian cancer reveals CD20, FoxP3 and TIA-1 as positive prognostic factors. PLoS One. 2009, 4(7), e6412
- Erbe, R.; Wang, Z.; Wu, S.; Xiu, J.; Zaidi, N.; La, J.; Tuck, D.; Fillmore, N.; Giraldo, N.A.; Topper, M.; Baylin, S.; Lippman, M.; Isaacs, C.; Basho, R.; Serebriiskii, I.; Lenz, H.J.; Astsaturov, I.; Marshall, J.; Taverna, J.; Lee, J.; Jaffee, E.M.; Roussos Torres, E.T.; Weeraratna, A.; Easwaran, H.; Fertig, E.J. Evaluating the impact of age on immune checkpoint therapy biomarkers. Cell Rep. 2021, 36(8), 109599
Reviewer 3 Report
The Authors sufficiently addressed reviewer's comment.
Author Response
Dear Reviewer
Thank you for reading our manuscript and critically reviewing it, which will help us improve it to a better scientific level and make it more understandable to the readership.
Round 3
Reviewer 2 Report
Authors made changes to discussion as suggested